# Targeted Quantification of Carbon Metabolites Identifies Metabolic Progression Markers and an Undiagnosed Case of SDH-Deficient Clear Cell Renal Cell Carcinoma in a German Cohort

**DOI:** 10.3390/metabo11110764

**Published:** 2021-11-09

**Authors:** Doreen William, Kati Erdmann, Jonas Ottemöller, Anastasios Mangelis, Catleen Conrad, Mirko Peitzsch, Evelin Schröck, Graeme Eisenhofer, Aristeidis Zacharis, Susanne Füssel, Daniela Aust, Barbara Klink, Susan Richter

**Affiliations:** 1Core Unit for Molecular Tumor Diagnostics (CMTD), National Center for Tumor Diseases Partner Site Dresden (NCT/UCC), 01307 Dresden, Germany; Doreen.William@nct-dresden.de (D.W.); Evelin.Schroeck@uniklinikum-dresden.de (E.S.); Daniela.Aust@uniklinikum-dresden.de (D.A.); Barbara.Klink@lns.etat.lu (B.K.); 2German Cancer Consortium (DKTK), 01307 Dresden, Germany; Kati.Erdmann@uniklinikum-dresden.de (K.E.); Susanne.Fuessel@uniklinikum-dresden.de (S.F.); 3German Cancer Research Center (DKFZ), 69120 Heidelberg, Germany; 4Department of Urology, Technische Universität Dresden, 01307 Dresden, Germany; Jonas.Ottemoeller@uniklinikum-dresden.de (J.O.); Aristeidis.Zacharis@uniklinikum-dresden.de (A.Z.); 5National Center for Tumor Diseases Partner Site Dresden (NCT/UCC), 01307 Dresden, Germany; 6School of Cardiovascular Medicine and Sciences, Faculty of Life Sciences and Medicine, King’s College London, London SE1 9NH, UK; anastasios.mangelis@kcl.ac.uk; 7Institute of Clinical Chemistry and Laboratory Medicine, Medical Faculty Carl Gustav Carus, University Hospital Carl Gustav Carus, Technische Universität Dresden, 01307 Dresden, Germany; Catleen.Conrad@uniklinikum-dresden.de (C.C.); Mirko.Peitzsch@uniklinikum-dresden.de (M.P.); Graeme.Eisenhofer@uniklinikum-dresden.de (G.E.); 8Institute for Clinical Genetics, University Hospital Carl Gustav Carus, Technische Universität Dresden, 01307 Dresden, Germany; 9ERN-GENTURIS, Hereditary Cancer Syndrome Center, 01307 Dresden, Germany; 10Department of Medicine III, University Hospital Dresden, 01307 Dresden, Germany; 11Institute of Pathology, Tumor and Normal Tissue Bank of the NCT/UCC Dresden, University Hospital Carl Gustav Carus, Technische Universität Dresden, 01307 Dresden, Germany; 12National Center of Genetics (NCG), Laboratoire National de Santé (LNS), 1, Rue Louis Rech, L-3555 Dudelange, Luxembourg

**Keywords:** Krebs cycle, metabolic profiling, renal cell carcinoma, subtypes, survival analysis, succinate dehydrogenase mutations

## Abstract

Renal cell carcinoma (RCC) is among the 10 most common cancer entities and can be categorised into distinct subtypes by differential expression of Krebs cycle genes. We investigated the predictive value of several targeted metabolites with regards to tumour stages and patient survival in an unselected cohort of 420 RCCs. Unsupervised hierarchical clustering of metabolite ratios identified two main clusters separated by α-ketoglutarate (α-KG) levels and sub-clusters with differential levels of the oncometabolite 2-hydroxyglutarate (2HG). Sub-clusters characterised by high 2HG were enriched in higher tumour stages, suggesting metabolite profiles might be suitable predictors of tumour stage or survival. Bootstrap forest models based on single metabolite signatures showed that lactate, 2HG, citrate, aspartate, asparagine, and glutamine better predicted the cancer-specific survival (CSS) of clear cell RCC patients, whereas succinate and α-ketoglutarate were better CSS predictors for papillary RCC patients. Additionally, this assay identifies rare cases of tumours with *SDHx* mutations, which are caused predominantly by germline mutations and which predispose to development of different neoplasms. Hence, analysis of selected metabolites should be further evaluated for potential utility in liquid biopsies, which can be obtained using less invasive methods and potentially facilitate disease monitoring for both patients and caregivers.

## 1. Introduction

The World Health Organization (WHO) classifies renal cell tumours into 16 subgroups, with clear cell RCC (ccRCC) being the most common, accounting for 80% of malignant RCCs [1]. The papillary RCC (papRCC) and chromophobe (chrRCC) histological subtypes make up most of the remaining cases. Rare forms of RCC can arise due to loss-of-function of fumarate hydratase (FH) or succinate dehydrogenase (SDH), mostly caused by germline mutations in those genes. Prevalence of both subtypes were estimated below 1% by immunohistochemistry [2]. SDH-deficient kidney tumours often present as oncocytomas, with a morphology of an eosinophilic and flocculent cytoplasm, whereas FH-deficient RCCs are histologically grouped as papRCC type II [3,4].

Metabolic reprogramming is an emerging hallmark of cancer and was previously characterised in RCC on a transcriptomic level [5]. Expression of Krebs cycle and electron transport chain genes was able to separate histological subtypes, with expression in ccRCC being lower than in other groups [6]. Metabolomics investigates changes on the functional level of metabolites and can be applied in an untargeted approach or with targeted assays that provide higher sensitivities for compound detection. Several studies have used untargeted metabolomics in tissue, serum, or urine to identify signatures for RCC subtypes and markers differentiating benign from cancerous lesions [7,8,9,10,11,12]. Some of these studies also evaluated metabolic changes occurring with an increased tumour stage. Citrate and other Krebs cycle intermediates, fatty acids, and lipid metabolism were identified to be disturbed in higher RCC stages [7,8,12]. Additionally, Poplawski et al. identified metabolites related to decreased patient survival, which include succinate, purines, glucose, beta-alanine, and myo-inositol [10].

We investigated whether a targeted set of metabolites containing small organic acids of the Krebs cycle, lactate, and the amino acids aspartate, asparagine, and glutamine, could be used to identify advanced tumour stages and to predict patient survival. Additionally, our method functioned as a screening tool for tumours caused by mutations in Krebs cycle genes. This approach, termed metabologenomics, was already successfully used to identify SDH- and FH-deficient phaeochromocytoma and paraganglioma [13].

## 2. Results

### 2.1. Metabolic Profiling of 420 RCCs Identified Two Main Clusters Separated by α-KG Levels and Two Sub-Clusters with Differential Concentrations of 2HG

Metabolite profiles of 13 small organic acids were generated for 420 RCCs (Table 1). Lactate and pyruvate were not quantifiable in all samples due to technical challenges, resulting in 318 tumours with complete profiles. A Spearman correlation analysis of the metabolites showed that cis-aconitate and isocitrate strongly correlated with citrate (ρ = 0.74 and 0.75) and with each other (ρ = 0.86) and that malate and fumarate were closely linked (ρ = 0.94, Appendix A). Therefore, cis-aconitate, isocitrate, and malate were excluded from further statistical analysis. Age and sex were weakly correlated with metabolite levels according to Spearman’s rank correlation coefficients. Significance was detected for three metabolites (glutamine, fumarate, and cis-aconitate) in relation to sex and, for asparagine, in relation to age (ρ = 0.14).

Unsupervised hierarchical clustering of all possible metabolite ratios resulted in two main groups separated by α-KG levels (Figure 1), with the majority of non-ccRCCs located in the low α-KG cluster (cluster B). Each group contained two sub-clusters based on differential levels of 2HG. Sub-clusters characterised by high 2HG were enriched in higher tumour stages (clusters A1 and B1), suggesting metabolite profiles might be suitable predictors of tumour stage or survival.

### 2.2. Lactate, 2HG, Succinate, and Citrate Are Independent Predictors of Tumour Stage and CSS

To explore the relationship between metabolite levels and tumour stage further, we evaluated which metabolites were significantly different between lower (pT1/2) and higher (pT3/4) primary tumour classification and TNM stage (Appendix A). Additionally, we compared metabolite ratios between necrotic and non-necrotic samples (Appendix A), as necrosis can influence metabolites in RCC tissues [11,12]. Altered metabolites were similar for these two comparisons (the one between tumour stages and the one between necrotic and non-necrotic samples): 2HG, succinate, lactate, and asparagine were elevated, whereas citrate, aspartate, and glutamine were lower in high-stage versus low-stage and in necrotic versus non-necrotic samples. Fumarate was decreased in necrotic tissue but was unaffected by tumour stage. α-KG levels were lower in higher stage tumours but did not significantly change in necrotic samples.

Since necrosis was more common in samples of higher primary tumour classification (Appendix A), we analysed metabolic changes excluding the influence of necrosis. Differences between high and low TNM stage were highly significant for 2HG, citrate, succinate, and lactate, followed by aspartate, asparagine, and glutamine (Figure 2). α-KG did not show significance after excluding necrotic samples. Repeating unsupervised hierarchical clustering without necrotic samples revealed a cluster with high succinate levels that was enriched in subtypes other than ccRCC (Appendix A)

2HG, citrate, succinate, and lactate were analysed for their predictive value of TNM stage. ROC curves for single metabolites showed areas under the curve (AUC) of 0.68 for citrate to 0.75 for lactate (Figure 3A). Using a bootstrap forest model to predict high or low risk tumours based on metabolite measurements increased the AUC to 0.90 (Figure 3B). Combining the different metabolites did not increase predictive power (Figure 3C). Validation of the bootstrap forest models is summarised in Appendix A. Metabolite-based risk predictions were reflected by CSS (Appendix A). Patients with higher risk tumours had significantly lower CSS distribution rates compared to patients with lower risk tumours based on tissue lactate (log rank *p* < 0.0001), succinate (*p* = 0.009), 2HG (*p* = 0.007), and citrate (*p* = 0.008) measurements. Using a bootstrap forest model for the other metabolites resulted in similar AUCs for prediction of risk, with significantly different CSS distributions between metabolite-based risk classifications for glutamine (*p* = 0.0005), aspartate (*p* = 0.002), and asparagine (*p* = 0.005), but not for α-KG (Appendix A).

### 2.3. Differences in Prognostic Metabolites between Patients with ccRCC and papRCC

When RCC type was plotted separately, it became apparent that lactate, 2HG, citrate, aspartate, asparagine, and glutamine better predicted the survival of patients with ccRCC, whereas succinate and α-KG were better CSS predictors for patients with papRCC (Figure 4).

Succinate- and α-KG-based risk predictions significantly separated high and low risk papRCC (log rank *p* = 0.019 and *p* = 0.0004, respectively). This was also true when necrotic samples were excluded from the survival analysis (data not shown). Type 2 papRCC were shown to have a worse prognosis than type 1 [14]; however, in our cohort, only a trend of higher succinate levels in type 2 papRCC was observed, whereas α-KG was similar (Appendix A). None of the other metabolite-based risk predictions reached statistical significance for papRCC, but they showed significance in predicting CSS in ccRCC by a log rank test and Cox proportional hazard model (Figure 4).

### 2.4. Metabologenomics Identified a Previously Undiagnosed Case of SDH-Deficient RCC

Metabolic screening identified 22 RCCs with either elevated ratios of succinate:fumarate (*n* = 12, Figure 5A) and α-KG:citrate (*n* = 1, Figure 5B) or high levels of 2HG (*n* = 9, Figure 5C, Appendix A). Elevated 2HG was explained by elevations of the L-enantiomer (Figure 5D). Sequencing data were available for four additional RCC cases with unremarkable metabolite profiles. The tumours were sequenced with a custom NGS Panel covering 84 cancer- and metabolism-associated genes to identify underlying genetic alterations [15]. The majority of RCCs showed pathogenic variants in VHL (17/26 cases, 65.4%; Figure 5E, Appendix A) independent of their metabolic profile. In RCCs with high levels of 2HG and the tumour with an elevated α-KG:citrate ratio, no underlying genetic alterations were identified by panel sequencing.

The RCC with the highest succinate:fumarate ratio (case RCC_163) showed a missense variant in SDHB (p.Leu65Pro, rs876659329). According to ACMG-AMP criteria, this variant was classified as likely pathogenic. In addition to the missense variant, RCC_163 showed deletion of one SDHB allele (Figure 5F). We analysed protein levels of respiratory chain complexes, including SDHB in six tumours with a comparable tumour cell content. SDHB staining was absent in RCC_163, but it was detectable in all other samples. Expression of respiratory chain complexes is known to be decreased in RCCs with loss of function alterations in VHL [16]. All but two analysed samples (RCC_163 and RCC_268) had such a loss of function mutation in VHL and showed lower levels of respiratory chain complexes II–IV, whereas complex V was similarly expressed in all samples (Figure 5G). For RCC_268, no underlying genetic aberration was found (Appendix A).

## 3. Discussion

Metabolic adaptions in cancer cells have long been recognised as an important part of carcinogenesis and tumour progression [5]. In this study, we showed that Krebs cycle metabolites and related amino acids carry a strong predictive value for tumour staging and that a subset of these metabolites is prognostic for patient survival. Interestingly, we identified RCC subtype-dependent differences in respect to predicting CSS.

In our initial unsupervised clustering analysis including necrotic tissues, two main clusters of RCCs, divided by relative levels of α-KG, were identified. Further analyses showed that this division was based on a higher number of necrotic samples in the higher stage tumours. α-KG dehydrogenase and also other genes of the Krebs cycle were shown to be differentially expressed in the TCGA cohort of RCC samples, in which higher expression was associated with a survival benefit [17]. Targeted analysis of α-KG in our study did not show a correlation with CSS in ccRCC patients, but it significantly predicted CSS in papRCC patients, also when necrotic samples were excluded from the analysis. This highlights the profound differences between RCC subtypes and the need for subtype-specific survival predictions.

The second metabolite that globally divided the tumours in our RCC cohort was 2HG. This is consistent with a report that identified L-2HG, particularly, as an epigenetic modifier, which was formed due to low expression levels of L2HGDH, the enzyme converting L-2HG to α-KG [18]. Our study confirmed that tumours with moderate elevations in 2HG mainly produce the L-enantiomer. Consistent with findings from untargeted metabolomics studies [12], we also showed that 2HG levels correlated with tumour stage and predicted CSS.

From our set of metabolites, 2HG, lactate, citrate, and succinate correlated with tumour stage, independent of metabolic changes due to necrosis, and they were predictors of CSS. A previous study showed citrate to decrease and succinate to increase with higher tumour stages [8]. Additionally, it was reported that succinate was associated with a worse outcome; however, patient numbers were low, with only 35 ccRCC [10]. In our cohort, citrate predicted survival in ccRCC and succinate better predicted survival of patients with papRCC. The latter could be associated with differences between type 1 and type 2 papRCCs [14]. Due to low numbers of papRCCs, we cannot conclusively evaluate this question.

Lactate elevations are characteristic for cells with a shift towards glycolytic metabolism, the so-called Warburg effect, which was shown in ccRCCs compared to non-transformed tissues and in 13C tracing studies in ccRCC tumours [8,19,20]. The majority of ccRCCs carry inactivating mutations in the VHL gene, causing stabilisation of hypoxia-inducible factors and thereby enhancing glycolysis [21]. Our results concerning the prognostic value of lactate are consistent with previous findings that LDHA expression in tumour tissue and serum lactate dehydrogenase from ccRCC patients are associated with survival [22,23].

In addition to the predictive value for tumour staging and patient survival, metabolite profiles aid in the identification of patients with hereditary cancer syndromes and more aggressive forms of RCC. A recent study in patients with advanced RCC demonstrated the occurrence of pathogenic germline mutations in genes predisposing to cancer development in 16.1% of cases, with 5.5% carrying aberrations in syndromic RCC-associated genes, most commonly *FH* and *SDHx* [24]. Loss of function in both genes can be detected by metabolic screening, as SDH-deficiency leads to accumulation of succinate and FH-deficiency to accumulation of fumarate, as our group has previously shown for phaeochromocytoma/paraganglioma [25]. Both SDH-deficient and FH-deficient RCCs are associated with a poor prognosis and warrant germline analysis to identify patients with hereditary cancer syndromes [26]. SDH-deficient RCCs are a distinct group according to WHO classification, and patients carry a higher risk of developing metastatic disease and should be carefully monitored [26,27]. One case of previously undiagnosed SDHB-mutated RCC was identified in our cohort by metabolic profiling and genetic analysis.

The presented study is a retrospective analysis, for which only tissue samples were available. For further analyses, it would be of interest to test the prognostic value of metabolites in liquid biopsies, since these are easily available pre-operatively. One publication investigated serum and urine metabolites in a small number of RCC patients and identified, amongst others, citrate, lactate, and succinate to be correlated with tumour stage [9], although succinate in urine was decreased in higher stages, not increased as we found in tumour tissue. Another shortcoming of this study is the low numbers of the less common papRCCs and chromophobe RCCs, making subtype stratification difficult.

Our analysis demonstrated that profiling a subset of intermediates from the central carbon metabolism in RCC could be clinically utilised in two ways. On the one hand, elevations of succinate, but also fumarate, can lead to the identification of previously unknown hereditary syndromes that have not only implications for the patient, but potentially also family members. On the other hand, we have shown that some metabolites are better predictive markers for ccRRC and others for papRCC, reflecting the different genetic background and metabolic rewiring happening in these subtypes. Subtype-specific differences should be investigated in more depth in the future to generate better models for outcome prediction and improve patient stratification.

## 4. Materials and Methods

### 4.1. Patient Cohort and Sample Collection

RCC specimens from patients treated by partial or total nephrectomy at the Department of Urology (Dresden, Germany) between 2010 and 2016 were collected and stored freshly at the Tumour and Normal Tissue Bank of the NCT/UCC, Dresden. Patients signed informed consent, approved by the local ethics committee. The histological RCC subtype, tumour tissue content, and necrosis were scored on haematoxylin/eosin sections. The Union for International Cancer Control (UICC) tumour, node, and metastasis (TNM) staging system was used to evaluate the progression state of the cancers (*n* = 420). Survival data were available for 266 patients. The median follow up was 50.9 months, ranging from 0.6 to 129.9 months. The pathological classification is summarised in Table 1.

### 4.2. Metabolite Profile

Frozen tissue (5–10 mg) was homogenised in methanol and analysed by high performance liquid chromatography tandem-mass spectrometry (LC-MS/MS), as described elsewhere. [25]. In addition, metabolite extracts were further analysed for the D- and L-enantiomers of 2HG by LC-MS/MS, according to a previously described method [13].

### 4.3. Genetic Testing and Variant Classification

Selected samples were analysed by next generation sequencing (NGS) on a NextSeq 500 sequencer (Illumina Inc., San Diego, CA, USA) using a custom designed multi-gene panel (Illumina Inc., San Diego, USA) [15]. Alignment of sequencing data to the human reference genome GRCh37 and variant calling were conducted using the CLC Genomics Workbench v20 (Qiagen, Hilden, Germany). Variants were classified in accordance with the standards and guidelines of the American College of Medical Genetics and Genomics and the Association for Molecular Pathology (ACMG-AMP) [28]. Copy number variations (CNVs) were called using the R package ‘panelcn.MOPS’ [29] (http://www.r-project.org/; accessed on 11 March 2019), using DNA isolated from blood of 10 healthy individuals as normal controls.

### 4.4. Western Blot

Proteins were separated by size via SDS-PAGE, transferred onto a nitrocellulose membrane (Amersham), and stained with a primary antibody mix against respiratory chain complexes (abcam, ab110411) and β-actin (Merck, MAB150R). HRP-conjugated anti-mouse antibodies (Sigma-Aldrich, 12–349) were used for detection.

### 4.5. Statistics

Evaluation of normality was done by a Shapiro–Wilk test. Since most metabolites followed non-normal distributions, non-parametric Mann–Whitney rank sum tests were used for comparisons (SigmaPlot version 12.5, Systat Software GmbH, Erkrath, Germany). For evaluating a ge and gender correlations to the metabolites, as well as in between metabolites, a Spearman’s rank correlation coefficient (ρ) was used.

Natural logarithm transformations were applied to all metabolite variables before hierarchical clustering, receiver operator characteristic (ROC) curve analysis, and bootstrap forest model generation. Bootstrap forest models were built for every metabolite using the high/low TNM stage classification as a response variable. The resulting metabolite-based risk prediction was used to investigate whether patient survival differs between risk groups. Internal validation was assessed by random dataset splitting (70/30%), using the weighted cross validation method.

CSS was used to evaluate the prognostic value of covariates using Kaplan–Meier curves and Cox proportional hazards models (hazard ratios, HR). Death from the disease was defined as the event, and time between first diagnosis and the date of death was defined as time to event. The prognostic value of covariates was tested by univariable Cox regression and log rank tests, and those with *p*-values less than 0.05 were considered significant.

All statistical tests and models were performed in JMP Pro version 15 (SAS Institute, Cary, NC, USA). Internal validation was performed in the JMP Autovalidation add-in.

## 5. Conclusions

Targeted analysis of tissue metabolites encompassing small organic acids of the central metabolism shows prognostic value in patients with RCCs. Especially lactate; 2HG; citrate; succinate; and the amino acids aspartate, asparagine, and glutamine are predictive of patient survival, and they have potential for further utility in liquid biopsies. Future investigations should put a specific focus on the differences between subtypes. The same mass spectrometry-based assay has identified rare cases of tumours with *SDHx* and *FH* mutations, which are caused predominantly by germline mutations. Identification of these patients becomes important since the risk of developing other tumours, including gastrointestinal stromal tumours, leiomyomas, phaeochromocytomas, and paragangliomas, is increased.

## Figures and Tables

**Figure 1 metabolites-11-00764-f001:**
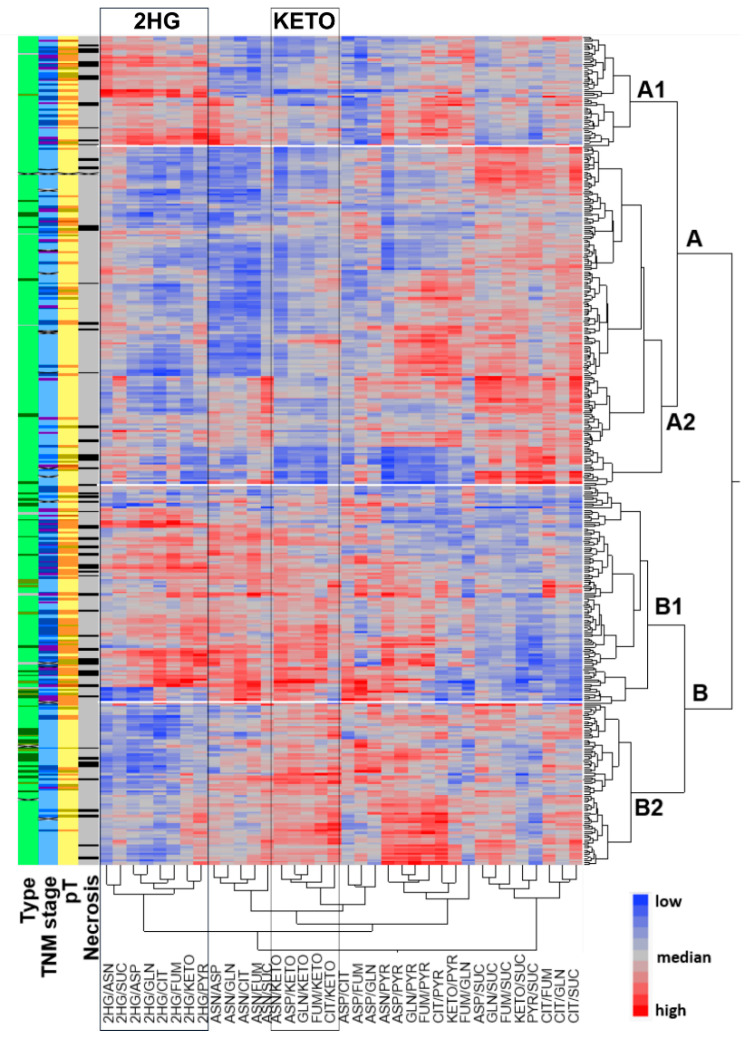
Unsupervised hierarchical clustering of metabolite ratios identified two main groups (A/B), predominantly based on differential α-ketoglutarate (KETO) levels. Each cluster contained two sub-clusters (1/2) characterised by differential 2HG levels. Colour coding: Necrosis—yes (black), no (grey); Primary tumour stage (pT) and TNM stage—darker is a higher stage; Type—CC (light green), P (dark green), CHR (olive), mixed type (grey).

**Figure 2 metabolites-11-00764-f002:**
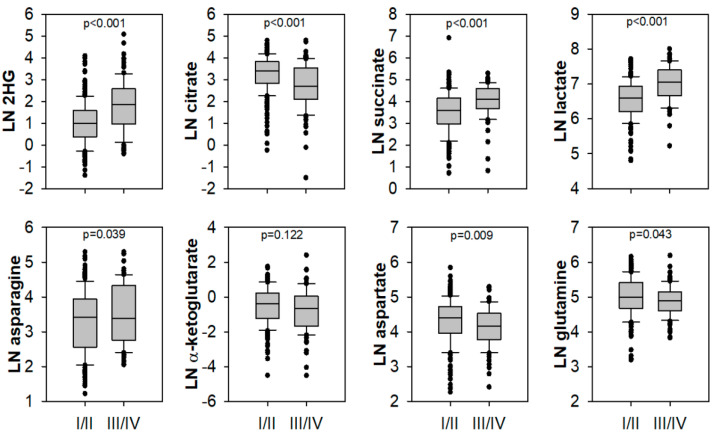
Significantly different metabolites between lower (I/II) and higher (III/IV) TNM stage RCC were independent of tumour necrosis. Only non-necrotic tumour samples were plotted, and significance was assessed by a Mann–Whitney U test, *n* = 342 (I/II 249; III/IV 93), except for pyruvate (*n* = 323) and lactate (*n* = 289).

**Figure 3 metabolites-11-00764-f003:**
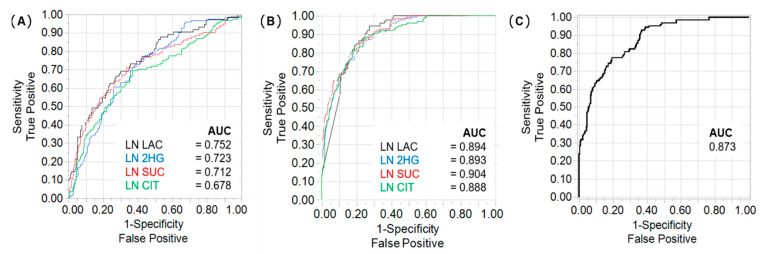
Metabolite signatures showed predictive value for high-risk tumours (TNM stage III and IV). (**A**) Receiver operator characteristic (ROC) curves for single metabolites (LN = logarithm) with areas under the curve (AUC) values; (**B**) Bootstrap forest model of single metabolites showed increased AUCs; sample set, *n* = 405; (**C**) Combination of metabolites did not improve the predictive power over single metabolites.

**Figure 4 metabolites-11-00764-f004:**
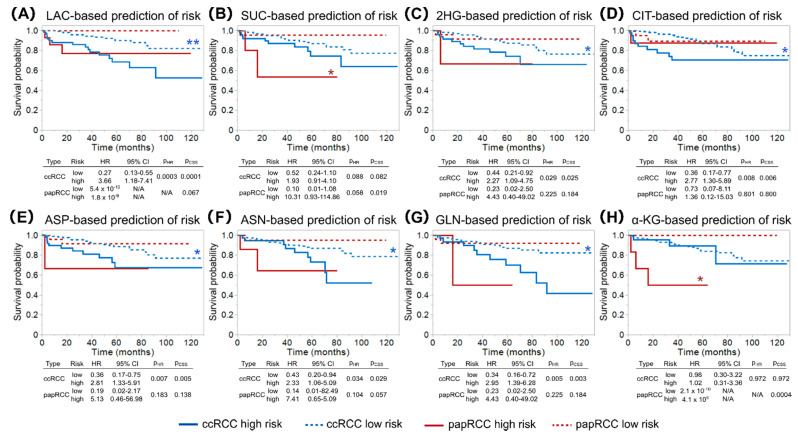
Survival analyses for risk predictions based on tumour metabolite levels of lactate (**A**), succinate (**B**), 2HG (**C**), citrate (**D**), aspartic acid (**E**), asparagine (**F**), glutamine (**G**), and α-KG (**H**). Metabolite-based predictions (low or high risk) were made by the bootstrap forest algorithm, and CSS was separately analysed for ccRCC (*n* = 227) and papRCC (*n* = 28) according to Kaplan–Meier (graphs) and Cox proportional hazard models (tables, HR—hazard ratios). Statistical difference was assessed by a log rank test for Kaplan–Meier analyses (pCSS) and a Wald test for Cox proportional hazards models (pHR). Significance was considered at * *p* < 0.05 and ** *p* < 0.001.

**Figure 5 metabolites-11-00764-f005:**
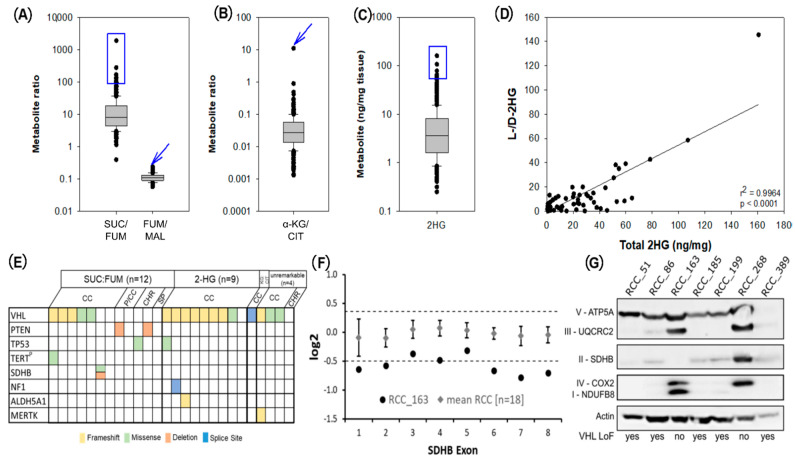
Screening for metabolic outliers with potentially underlying mutations using LC-MS/MS measurements of Krebs cycle metabolites. (**A**) Succinate:fumarate (SUC/FUM) and fumarate:malate (FUM/MAL) ratios; 12 tumours with the highest succinate:fumarate ratios and the tumour with the highest fumarate:malate ratios were sequenced (blue box and arrow, Appendix A). (**B**) α-ketoglutarate:citrate (α-KG/CIT) values (blue arrow, Appendix A). (**C**) Total 2-hydroxyglutarate (2HG) values; 10 tumours with the highest values were sequenced (blue box, see Appendix A). (**D**) In a subset of 51 tumours (31 samples with the highest total 2HG and 20 others), D and L enantiomers of 2HG were additionally measured to identify potential IDHx mutations. The correlation between total 2HG and the ratio of the L to D enantiomer (L-/D-2HG) indicated that increases in 2HG were due to an elevated L-enantiomer. (**E**) Oncoplot of RCC samples analysed by NGS. (**F**) CNV calling for all exons of SDHB using NGS data represented as log2 ratios of RCC_163 (black dots) and mean log2 ratios of 18 RCCs (grey squares), with standard deviation in relation to a pool of 10 normal controls. (**G**) Western blot analysis of respiratory complexes, with 20µg protein per sample; in the case of RCC_389, only 10 µg protein could be analysed due to sample limitations. Samples with loss of function (LoF) mutations in VHL are marked on the bottom of the plot. Abbreviations: SUC:FUM—succinate:fumarate ratio, KG:CIT—α-KG:citrate ratio, CC—clear cell RCC, P/CC—mixed papillary and clear cell RCC, CHR—chromophobe RCC, SP—mucinous tubular and spindle cell carcinoma.

**Table 1 metabolites-11-00764-t001:** Patient cohort with histological tumour characterization.

Feature	Details
Age at resection	68 (27–90) years ^1^
Sex	36% females
Tumour type	360 clear cell; 38 papillary (type 1 and 2); 6 mixed clear cell/papillary; 13 chromophobe; 1 mixed chromophobe/clear cell; 1 mucinous tubular/spindle cell; 1 unknown
Primary tumour stage	266 pT1; 40 pT2; 105 pT3; 8 pT4; 1 unknown
TNM stage	247 stage I; 33 stage II; 81 stage III; 44 stage IV; 15 unknown
Necrosis	16.0% (67/419, 1 unknown)

^1^ median and range in brackets.

## Data Availability

The data presented in this study are available on request from the corresponding author. The data are not publicly available due to reasons of data protection.

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
