# Peer review of "Targeted Quantification of Carbon Metabolites Identifies Metabolic Progression Markers and an Undiagnosed Case of SDH-Deficient Clear Cell Renal Cell Carcinoma in a German Cohort"

_metabolites, 2021, doi:10.3390/metabo11110764_

Round 1
Reviewer 1 Report
A well written research paper that focused on the predictive and prognostic value of several targeted metabolites in an unselected cohort of 420 RCCs. The results can be a starting point for analysis of selected metabolites in liquid biopsies.
Author Response
We thank the reviewer for the kind comments and the time spent reviewing the manuscript.
Reviewer 2 Report
In this manuscript, William et al report the predictive value of a set of metabolites in human renal cell carcinoma specimens (unselected cohort of 420 patients). They actually show that differential levels of alpha-ketoglutarate and 2-hydroxyglutarate allow to identify several (sub)clusters correlated with tumour stage. They also observe that survival of two RCC subtypes, namely clear cell and papillary, is predicted by the levels of distinct metabolites. Finally, in their study, a genomic analysis of tumor specimens has allowed the identification of rare cases of RCC harbouring SDHx mutations. Overall, the study is well designed and data are clearly presented. Conclusions are appropriate and well discussed. I have thus only minor comments to improve the clarity of the study:
1-In Table 1, the authors should also indicate the gender of the patients. They must also discuss whether there is a gender-dependent effect on metabolite levels in tumour specimens.
2-It is unclear for the reviewer why lactate and pyruvate levels are not quantifiable in all samples (318/420). Is it biologically relevant or due to any technical issue/limitation ? Please clarify.
3-Metabolite levels have been assessed from crude extracts of tumours. The tumour composition is likely different among the RCC subtypes as well as along disease progression. Do you have any information about the cellular composition in the different RCC subtypes and/or can you discuss about how it may affect the metabolomic profile in the tumour specimens ?
4-While certainly out of the scope of the current manuscript, I am wondering whether the authors have any evidence for epigenetic changes in their different tumour specimens/clusters. Indeed, as briefly indicated in line 213, 2HG is a well known epigenetic modulator. Any data for 5(h)mC levels in the samples ?
Author Response
1-In Table 1, the authors should also indicate the gender of the patients. They must also discuss whether there is a gender-dependent effect on metabolite levels in tumour specimens.
Reply: Thank you very much for this suggestion. To address the comment, we have added sex to Table 1, and a small description of the data was added to the Results part (line 90ff):”Age and sex were weakly correlated with metabolite levels according to Spearman’s rank correlation coefficients. Significance was detected for three metabolites (glutamine, fumarate, cis-aconitate) in relation to sex and for asparagine in relation to age (ρ=0.14)”. Due to lack of strong indications, we will refrain from discussing sex-related influences in the main text. Furthermore, an additional line was added to the statistical analysis paragraph in the methods section: “For evaluating age and gender correlations with the metabolites, as well as in between metabolites, Spearman’s rank correlation coefficient (ρ) was used.”
2-It is unclear for the reviewer why lactate and pyruvate levels are not quantifiable in all samples (318/420). Is it biologically relevant or due to any technical issue/limitation ? Please clarify.
Reply: This has now been clarified in the text. Line 85: Lactate and pyruvate were not quantifiable in all samples due to technical challenges, resulting in 318 tumours with complete profiles.
3-Metabolite levels have been assessed from crude extracts of tumours. The tumour composition is likely different among the RCC subtypes as well as along disease progression. Do you have any information about the cellular composition in the different RCC subtypes and/or can you discuss about how it may affect the metabolomic profile in the tumour specimens ?
Reply: Thank you very much for this question. Tissue heterogeneity is definitely an issue. For this reason, we have evaluated the tumour cell content in every tissue block and discarded samples with tumour cell contents below 50%. We have also checked whether the distribution of tumour cell content was different between subtypes and tumour stages, which was not the case. We have not evaluated the tissue composition beyond the tumour cell content.
The reviewer is correct, in that different cell types contain different levels of metabolites, and that a dilution of the tumour cells by stromal and immune cells will change the metabolic profiles.
4-While certainly out of the scope of the current manuscript, I am wondering whether the authors have any evidence for epigenetic changes in their different tumour specimens/clusters. Indeed, as briefly indicated in line 213, 2HG is a well known epigenetic modulator. Any data for 5(h)mC levels in the samples ?
Reply: This is a very interesting point that the reviewer raises, and is definitely worth following up on in the future. Unfortunately, we have not collected data on DNA methylation or 5(h)mC levels.
Reviewer 3 Report
The authors using targeted quantification of metabolites screened about 420 patients with RCC in order to distinguish the subtype of RCC and predict their progression. The cohort of patients is statistically relevant however some of the findings are not novel. Especially the results of 2-HG were already described (Hakimi AA, Cancer Cell 2016; Shim EH Cancer Discovery 2014) together with results on lactate (Courtney KD, Cell Metabolism 2018; Girgis H, Molecular Cancer 2014).
Major comments:
Why did the authors decide to measure only these 13 metabolites? Do they have results also on other aminoacids?
Unsupervised hierarchical clustering , Figure 1 is based mainly on differential levels of a-KG however in the next paragraph authors show that a-KG levels are affected by necrotic samples. It would be important to perform HCA again without necrotic samples. Next, authors claim that levels of a-KG in necrotic samples were lower in higher stage tumors. When looking at supplementary figure 3 and 4 it is not clear. They should show T3/T4 samples only and divide them to non-necrotic and necrotic and compare the levels of the metabolites.
Regarding the prediction I would appreciate if you better explain the statistical analysis used since it seems that first you divided the patients according the bootstrap forest model to high and low risk tumors based on metabolite levels and then you just show by survival probability graph, supplementary Fig.6 that the high stage tumors survive worse which is expected. Maybe it is only the matter of missing description of the whole statistical analysis. Also what is on the y axis under survival probability, is it death or also relapse? I would also expect to show the prediction value on validation cohort.
In the discussion I am missing conclusion based on observed findings, how can be these results applied in the clinics or in the understanding of the nature of the cancer. Why these specific metabolites are distinct among the studied groups and also you dont discuss the significance of your findings on the predictive value of the metabolites.
Author Response
The authors using targeted quantification of metabolites screened about 420 patients with RCC in order to distinguish the subtype of RCC and predict their progression. The cohort of patients is statistically relevant however some of the findings are not novel. Especially the results of 2-HG were already described (Hakimi AA, Cancer Cell 2016; Shim EH Cancer Discovery 2014) together with results on lactate (Courtney KD, Cell Metabolism 2018; Girgis H, Molecular Cancer 2014).
Reply: Thank you very much for taking the time to review our manuscript. We fully understand your concerns, but we would like to point out that our study provides confirmatory data for previous studies (e.g. those mentioned above) and also presents new results about the prognostic ability of different metabolites, especially in the context of papillary versus clear cell RCC.
Major comments:
Why did the authors decide to measure only these 13 metabolites? Do they have results also on other aminoacids?
Reply: We have chosen an already available analytical method in our laboratory that focusses on acids of the Krebs cycle. We have used this assay in the past to identify tumours with mutations in Krebs cycle genes. Since this was also an objective of the presented study, we have chosen the assay. Other amino acids or sugar metabolites are not part of this method.
Unsupervised hierarchical clustering , Figure 1 is based mainly on differential levels of a-KG however in the next paragraph authors show that a-KG levels are affected by necrotic samples. It would be important to perform HCA again without necrotic samples. Next, authors claim that levels of a-KG in necrotic samples were lower in higher stage tumors. When looking at supplementary figure 3 and 4 it is not clear. They should show T3/T4 samples only and divide them to non-necrotic and necrotic and compare the levels of the metabolites.
Reply: Thank you for this suggestion. We have now added the hierarchical clustering excluding necrotic samples to the supplement, (new Supplementary Figure S6), and added to the results (line123ff).
Apologies for the confusion concerning a-KG in necrotic samples. The paragraph describing this data is not very clear. In fact, we did not intent to claim what the reviewer is detailing above. This paragraph (line111ff) has now been rewritten, as well as the part in the discussion (line 217ff).
Regarding the prediction I would appreciate if you better explain the statistical analysis used since it seems that first you divided the patients according the bootstrap forest model to high and low risk tumors based on metabolite levels and then you just show by survival probability graph, supplementary Fig.6 that the high stage tumors survive worse which is expected. Maybe it is only the matter of missing description of the whole statistical analysis. Also what is on the y axis under survival probability, is it death or also relapse? I would also expect to show the prediction value on validation cohort.
Reply: Thank you for raising this issue. We have clarified the method as follows (line 313ff): “Bootstrap forest models were built for every metabolite using the high/low TNM stage classification as response variable. The resulting metabolite-based risk prediction was used to investigate whether patient survival differs between risk groups.”
The y-axis of Figure S8 (old Figure S6) shows disease-specific survival, i.e. deaths, and does not include relapse. This has now been clarified in the figure legend: “Survival analyses for risk assessment based on tumour metabolite levels of lactate (A), succinate (B), 2HG (C), and citrate (D). The figures show probability of disease-specific deaths (which does not include relapse) over time. Metabolite-based predictions were made by the bootstrap forest algorithm, and CSS (n=266) was plotted. Univariable Cox proportional hazards models were used for calculation of hazard ratios (HR) with significance levels at *p<0.05 and **p<0.001.”
Validation of the bootstrap forest model was added to the text (line 135ff) together with a new Supplementary Figure S7. Additionally the procedure was added to the methods section (line 316ff and 323ff): “Internal validation was assessed my random dataset splitting (70/30%) using weighted cross validation method.” and “Internal validation was performed in the JMP Autovalidation add-in.”
In the discussion I am missing conclusion based on observed findings, how can be these results applied in the clinics or in the understanding of the nature of the cancer. Why these specific metabolites are distinct among the studied groups and also you dont discuss the significance of your findings on the predictive value of the metabolites.
Reply: Thank you very much for pointing out this shortcoming. We have now added the following paragraph at the end to the discussion (line 265ff): “Our analysis demonstrated that profiling a subset of intermediates from the central carbon metabolism in RCC could be clinically utilised in two ways. On the one hand, elevations of succinate but also fumarate can lead to the identification of previously unknown hereditary syndromes that have not only implications for the patient, but potentially also family members. On the other hand, we show that some metabolites are better predictive markers for ccRRC and others for papRCC, reflecting the different genetic background and metabolic rewiring happening in these subtypes. The identified metabolites represent promising candidates to improve outcome prediction, and consequently, to optimise follow-up and treatment schedules for RCC patients. Patients with altered levels of these biomarkers should be closer monitored following nephrectomy. Sub-type-specific differences should be investigated in more depth in the future to generate better models for outcome prediction and improve patient stratification. Further studies are warranted to translate the tissue findings to liquid biopsies such as plasma, which would allow a minimal-invasive and longitudinal determination of biomarkers and thus, an easier implementation into clinical decision making.”
Round 2
Reviewer 3 Report
It is significant that the authors took into account and analysed the impact of necrosis in the samples. However, I think they should separate the results of non-necrotic and necrotic samples as they did it in Supplementary 7. I think the result from Supplementary should be used in the main text and the results of HCA Figure 1 should be in the Supplementary and the authors should rewrite their statement based on HCA data.
Also regarding Figure 2 and Figure S4, again they are showing the results of combined non-necrotic and necrotic samples. They should show the results excluding the necrotic samples.
The authors show that the results from HCA are different once they excluded necrotic samples. Therefore I don’t think the conclusion can be drawn on results from combined necrotic and non-necrotic samples. The authors should separate these results and change the conclusion otherwise the paper is not acceptable.
Author Response
Thank for taking the time to thoroughly review our manuscript. We agree that separating the necrotic samples is important to initially confirm the role of the different metabolites in respect to tumour stage and that is exactly why we took necrosis into account from the beginning.
In actual fact, Figure 2 shows the results “independent of tumour necrosis”. This was the case already in the first version of this manuscript. The figure legend was clarified and reads: “Significantly different metabolites between lower (I/II) and higher (III/IV) TNM stage RCC are independent of tumour necrosis. Only non-necrotic tumour samples were plotted and significance was assessed by Mann-Whitney U test, n = 342 (I/II 249; III/IV 93), except for pyruvate n = 323 and lactate n = 289.”
Since the purpose of our manuscript is to use metabolites as classifiers for patient survival, we disagree with exchanging Figure 1 with Supplementary Figure 6 (I assume this is what the reviewer refers to). Figure 1 described the situation in an unselected manner, and this means also necrotic samples carry information through their metabolic phenotype. This information is also incorporated into the machine learning approach to classify patients according to low or high risk. Since necrosis is common amongst RCCs, it would be a limitation of such an approach to exclude necrotic patients from the beginning.
It was an excellent suggestion from the reviewer to repeat hierarchical clustering without necrotic samples, which better indicated the importance of succinate in the group of non-RCC tumours. Later we also identified succinate by machine learning based classification to be important in papRCC. Similarly, we also identified a significant role of α-KG in papRCC in respect to patient survival, even though α-KG levels were not significantly changed in the overall cohort without necrotic samples. To further substantiate our analysis, we have repeated the survival analysis excluding necrotic samples, which gave the same results for α-KG and succinate in papRCC (as an example the results for α-KG are displayed below). We have added one sentence to the results part (line 164/165): “This was also true when necrotic samples were excluded from the survival analysis (data not shown).” In the discussion, we then rightfully state that “the low numbers of the less common papRCCs and chromophobe RCCs” is a “shortcoming of this study”, and requires further clarification.
Since the only metabolite that was not significantly changed after excluding necrotic samples was α-KG, I can only assume the reviewer refers to the conclusions about this one particular metabolite. For all others, we demonstrated that even after removing necrotic samples, significant difference between high and low tumour stage was reached (refer to Figure 2). The paragraph on α-KG in the discussion was modified as follows: “In our initial unsupervised clustering analysis including necrotic tissues, two main clusters of RCCs, divided by relative levels of α-KG were identified. Further analyses showed that this division was based on a higher number of necrotic samples in the higher stage tumours. α-KG dehydrogenase and also other genes of the Krebs cycle were shown to be differentially expressed in the TCGA cohort of RCC samples, in which higher expression was associated with a survival benefit [20]. Targeted analysis of α-KG in our study did not show correlation with CSS in ccRCC patients, but significantly predicted CSS in papRCC patients also when necrotic samples were excluded from the analysis. This highlights the profound differences between RCC subtypes and the need for subtype-specific survival predictions.”
Survival plots for α-KG excluding necrotic samples in ccRCC (left) and papRCC (right): see attached file

Round 3
Reviewer 3 Report
Thank you for clarification. I have no more comments.